# Multi-modal Collaborative Optimization and Expansion Network for Event-assisted Single-eye Expression Recognition

## Abstract

In this paper, we proposed a Multi-modal Collaborative Optimization and Expansion Network (MCO-E Net), to leverage event modalities to resist challenges such as low light, high exposure, and high dynamic range in single-eye expression recognition tasks. The MCO-E Net introduces two innovative designs: Multi-modal Collaborative Optimization Mamba (MCO-Mamba) and Heterogeneous Collaborative and Expansion Mixture-of-Experts (HCE-MoE). MCO-Mamba, building upon Mamba, leverages dual-modal information to jointly optimize the model, facilitating collaborative interaction and fusion of modal semantics. This approach encourages the model to balance the learning of both modalities and harness their respective strengths. HCE-MoE, on the other hand, employs a dynamic routing mechanism to distribute structurally varied experts (deep, attention, and focal), fostering collaborative learning of complementary semantics. This heterogeneous architecture systematically integrates diverse feature extraction paradigms to comprehensively capture expression semantics. Extensive experiments demonstrate that our proposed network achieves competitive performance in the task of single-eye expression recognition, especially under poor lighting conditions. Anonymous code repository is provided in Appendix A.

## 1 Introduction

Single-eye Expression Recognition analyzes eye movement patterns via visual sensors. This emerging technique offers privacy/occlusion advantages over facial recognition (Hickson et al., 2019; Barros & Sciutti, 2021; Wu et al., 2020a) and benefits applications like driver monitoring and HCI. However, illumination challenges (low-light/HDR/overexposure) degrade performance. While existing solutions use infrared (Wu et al., 2020a) or depth sensors (Lee et al., 2020a; Siddiqi et al., 2014), they fail to capture critical ocular texture details and micro-movements.

Event cameras capture spatiotemporal light changes (position, timing, polarity) through asynchronous outputs, excelling in extreme illumination scenarios via ultra-high temporal resolution to track subtle ocular dynamics. However, event streams exhibit extreme semantic sparsity compared to RGB data, limiting discriminative feature extraction. This inherent sparsity necessitates leveraging complementary RGB information, while fusing both modalities enhances expression semantics, significant challenges arise from fundamental differences in data generation mechanisms, spatiotemporal representations, and semantic richness, compounded by the complexity of modeling long-range temporal dependencies.

While SEEN (Zhang et al., 2023a) pioneered RGB–Event fusion through direct addition, and MSKD (Wang et al., 2024b) introduced cross-modal distillation for knowledge transfer between RGB and Event modalities, both approaches neglect fine-grained modality alignment. This simple static fusion mechanism ignores the deep semantic alignment problem between modalities, often leading to inconsistent semantic representations and limiting the representation ability of the model. To address this, we explore collaborative modeling of long-sequence Event and RGB data. Recently, Mamba-based Methods (Zhang et al., 2025; Dong et al., 2024b;a; Liu et al., 2024a) demonstrate superior performance in multimodal semantic collaborative perception and modeling. However, the direct concatenation of dual-modality semantics still leads to misalignment and inconsistencies in

spatiotemporal semantics. Moreover, some methods (Wan et al., 2024; Wang et al., 2024a; Ye et al., 2025) attempt to adopt an alternating optimization model parameter space to align the semantic distributions of the two modalities. This is because the ultimate objective of deep models is to fit or learn the distribution of data. Consequently, the outcome of efficient collaborative modeling between two modalities is that the two modal distributions learned by the model become aligned. However, the above mentioned methods (Wan et al., 2024; Wang et al., 2024a; Ye et al., 2025), the two modes of information remain independent and lack direct participation of modality data in the process of trying to learn the common distribution.

In addition, there are common and unique semantics in the discrimination of different expressions. The semantic discrimination provided by the eye region alone is very limited. Based on the traditional single-branch deep model, the unique semantics of different expressions are easily coupled with each other, thereby reducing the discriminative ability. Mixture of Experts (MoE) (Jacobs et al., 1991; Shazeer et al., 2017) is a model that combines multiple sub-model experts and a gating mechanism to perceive and encode diverse semantic representations from different perspectives, to alleviate the semantic coupling problem of traditional deep models. Conventional MoE implementations face inherent limitations due to their architecturally homogeneous experts with equivalent representational capacities. This structural uniformity induces overlapping feature learning across experts, which undermines their potential for specialization. Recently, HMoE (Wang et al., 2024a) reveals that such homogeneity constrains models' ability to address heterogeneous complexity demands. MFG-HMoE (Chen et al., 2025) introduces modular heterogeneity by grouping experts sharing internal structures and varying convolution kernel sizes between groups, this approach retains intrinsic structural uniformity within groups and restricts diversity to a single parameter dimension (kernel size). However, these methods often suffer from the homogeneity problem among experts, which means that the abilities of experts may overlap when processing different semantics, resulting in poor performance of the model when dealing with complex and diverse inputs. Motivated by the need to overcome these limitations and achieve deeper, more fundamental specialization, we exploit inherent structural heterogeneity within our Mixture-of-Experts framework. By designing experts with diverse architectural foundations, we enable the extraction of truly distinct expertise to capture the varied complexity and multimodal nature of real-world data.

To solve the above two issues, we proposed a Multi-modal Collaborative Optimization and Expansion Network (MCO-E Net). The MCO-E Net contains two novel designs: Multi-modal Collaborative Optimization Mamba (MCO-Mamba), Heterogeneous Collaborative and Expansion MoE (HCE-MoE). In MCO-Mamba, based on Mamba, we use two-modal information to jointly optimize the model, and perform collaborative interaction and fusion of modal semantics to drive the model to balance the learning of two-modal semantics and capture the advantages of both modalities. In the HCE-MoE, distributes structurally diversified experts (deep, attention and focal) through a dynamic routing mechanism, enabling collaborative learning of complementary semantics. This heterogeneous architecture systematically combines diverse feature expertise knowledge extraction paradigms to capture comprehensive expression semantics. Contributions of this work are as follows:

- We design the MCO-Mamba to better align and fuse the Event and RGB modalities in a collaborative manner.
- We design the HCE-MoE to enable collaborative learning of complementary visual representations.
- Extensive experiments demonstrate that our MCO-E Net achieves competitive performance in event-based single-eye expression recognition.

## 2 RELATED WORKS

**Expression Recognition.** Current facial expression recognition methods (Zheng et al., 2023; Lee et al., 2020b; Zhang et al., 2023b; Li et al., 2023; Xu et al., 2025) predominantly rely on RGB data but exhibit sensitivity to lighting variations and occlusions. Subsequently, MRAN (Lee et al., 2020b) processes synchronized color, depth, and thermal streams via spatiotemporal attention mechanisms, while DMD (Li et al., 2023) employs graph-based distillation units to optimize cross-modal integration. However, these methods face practical challenges such as privacy concerns and hardware constraints. To relieve these issues, Zhang et al. (2023a) and MSKD (Wang et al., 2024b) are eye

expression recognition methods that use event streams to protect privacy and resist the challenges of poor lighting conditions. Inspired by them (Zhang et al., 2023a; Wang et al., 2024b), we further design an efficient RGB and event modality collaborative modeling mechanism to mine and fuse the semantic advantages of the two modalities.

**Mamba Framework.** The recently introduced Mamba (Gu & Dao, 2023) architecture, integrating State Space Models (SSMs) (Gu et al., 2022; 2021b) from control theory (Basar, 2001), combines fast inference with linear sequence-length scaling, enabling efficient long-range dependency modeling. Its vision-specific variants (Zhu et al., 2024; Liu et al., 2024b; Li et al., 2024) and multimodal extensions (Zhang et al., 2025; Dong et al., 2024b;a; Liu et al., 2024a) demonstrate strengths in processing heterogeneous data (video, audio, language). However, existing methods rely on direct feature concatenation without addressing modality gaps. Recent works like Sigma (Wan et al., 2024) (partial SSM parameter exchange), MSFMamba (Gao et al., 2025) (full parameter exchange), and DepMamba (Ye et al., 2025) (selective parameter sharing) attempt cross-modal alignment but enforce static fusion strategies, risking modality-specific feature degradation or shared-specific imbalance. To resolve this, we propose an adaptive coupling mechanism that dynamically balances modality-shared and modality-specific features during interaction, enabling task-driven cross-modal fusion.

**Mixture-of-Experts (MoE).** The MoE developed by Jacobs et al. (1991), enables specialized components to autonomously process segmented data domains and then integrate them uniformly. On this basis, SMoE (Shazeer et al., 2017) proposed Sparsely-Gated Mixture-of-Experts, which employs a gating network for expert selection and proposes a Top-$K$ routing strategy, that is, selecting the $K$ experts with the highest probability. Zhou et al. (2022b) proposed expert choice routing, change the routing method from selecting top-k experts for each token to selecting top-k tokens for each expert. Hard MoE (Gross et al., 2017), employing a single decoding layer, demonstrates efficient trainability while achieving competitive performance on large-scale hashtag prediction benchmarks. HMoE (Wang et al., 2024a) solves the expert homogeneity problem by changing the parameter dimension size of the expert sub-network, but this modulation mechanism fundamentally retains the unified structure of the experts. MFG-HMoE (Chen et al., 2025) introduces modular heterogeneity by grouping experts with identical internal structures within each group and varying convolution kernel sizes between different groups for super-resolution tasks. Different from these prior works (Wang et al., 2024a; Chen et al., 2025; Wang & Liu, 2025), we exploit structural heterogeneity to design a Mixture-of-Experts that can extract different expertise through experts with different structures.

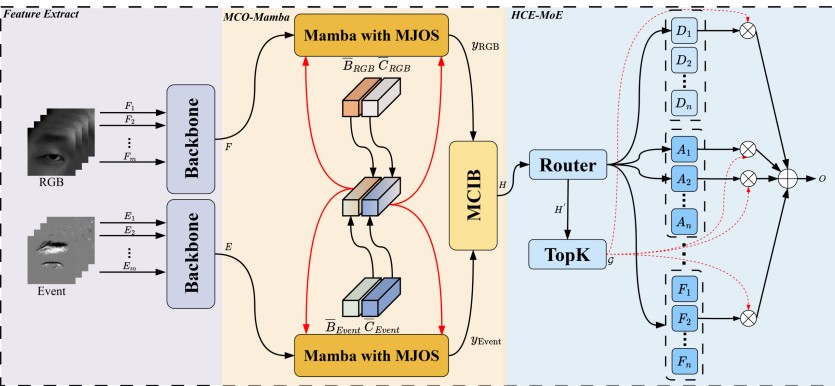

Figure 1: Overall architecture of our proposed MCO-E Net. The Network contains two novel designs: Multi-modal Collaborative Optimization Mamba (MCO-Mamba) and Heterogeneous Collaborative and Expansion Mixture of Experts (HCE-MoE).

## 3 METHODOLOGY

**Overview.** Overall architecture of our MCO-E Net is shown in Fig. 1. Input information contains: RGB sequences $F_i \in \mathbb{R}^{H \times W \times C}, i = 1, 2, \cdots, m$ and Event sequences $E_i \in \mathbb{R}^{H \times W \times B}, i =$

$1, 2, \cdots, m$, where $H, W, C$ and $B$ represent height, width, RGB channels and Event channels. **Firstly**, the RGB sequences and Event sequences are fed into their respective backbones; Each tensor is processed by ResNet-18 for feature extraction, and then all features are concatenated together to obtain $F$ and $E \in \mathbb{R}^{M \times E}$, $E$ is the feature dimension, $M$ is sequence length. **Next**, features $F$ and $E$ undergo our MCO-Mamba, jointly optimizing the model and performing collaborative interaction and fusion of modal semantics. **Finally**, the fused representation from MCO-Mamba is processed through our HCE-MoE. The HCE-MoE combines diverse feature expertise knowledge extraction paradigms to capture comprehensive expression semantics.

### 3.1 MCO-MAMBA

As mentioned in Intro. 1, the differences in information generation mechanism, representation form, and semantic richness between event streams and RGB data sequences lead to a modal semantic gap between the two modalities; In addition, both modalities exhibit long-sequence characteristics, which undoubtedly adds challenge to the collaborative modeling of modes. To relieve two sub-issues, we leverage the advantages of Mamba for long-sequence data modeling to design a multi-modal joint optimization and interactive representation model, to fully leverage the advantages of both modalities. To this end, we proposed the Multi-modal Collaborative Optimization Mamba (MCO-Mamba).

Our proposed MCO-Mamba is shown in Fig. 2, which consists of two core components: (i)Multi-modal Joint Optimization Scheme (MJOS) for Mamba: we jointly optimize Mamba with two modalities of information to improve its modeling of long-sequence information and perception of cross-modal information, thereby building a semantic bridge between the two modalities. (ii)Multi-modal Collaborative Interaction Block (MCIB): We further interactively represent and fuse the two modalities to achieve complementary advantages of the two modalities.

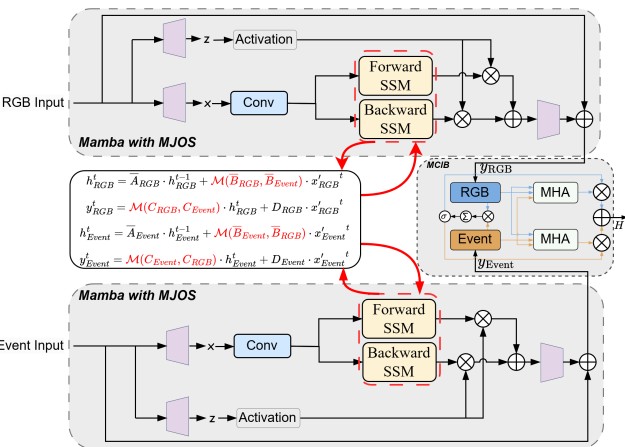

Figure 2: Architectures of proposed MCO-Mamba. We first jointly optimize the model using Event and RGB modalities to balance the learning of the two modal distributions; Next, we model the collaborative interaction between two modalities to leverage their respective strengths.

#### 3.1.1 MJOS FOR MAMBA.

To efficiently perceive and capture high-quality semantics from long-sequence data, we use Mamba to encode the semantics of Event streams and RGB sequences. However, Mamba is not good at collaborative modeling of two-modal semantics. Therefore, some methods (Zhang et al., 2025; Dong et al., 2024b;a; Liu et al., 2024a) integrate the semantics of Mamba after modeling, but this does not perceive the role of sequence information in the fusion of modal semantics. Other methods (Wan et al., 2024; Gao et al., 2025) are to alternately update the time series status information of Mamba. Although these methods can effectively perceive the temporal semantics of two modalities, they lack the joint participation of two-modal information. This is not conducive to the balance of two-mode semantic distribution and the effective capture of semantic meaning in deep models. So, we use two-modal sequence data to jointly optimize the state information of Mamba. Here, we use SSM

with bidirectional scanning (Zhu et al., 2024) to capture the forward and backward dependencies respectively to ensure that the output of each position can simultaneously refer to the context of the entire sequence. We define it as Multi-modal Joint Optimization Scheme (MJOS) for Mamba.

Specifically, details of our MJOS for Mamba are shown in Appendix Algorithm 1. At different times, we jointly optimize the state equation of Mamba using the RGB modality and Event modality (as shown in Fig. 2), as detailed below:

$$h_{\text{RGB}}^t = \overline{A}_{\text{RGB}} \cdot h_{\text{RGB}}^{t-1} + \underbrace{\mathcal{M}(\overline{B}_{\text{RGB}}, \overline{B}_{\text{Event}})}_{\text{Joint Optimization}} \cdot {x'_{\text{RGB}}}^t \tag{1}$$

$$y_{\text{RGB}}^t = \underbrace{\mathcal{M}(C_{\text{RGB}}, C_{\text{Event}})}_{\text{Joint Optimization}} \cdot h_{\text{RGB}}^t + D_{\text{RGB}} \cdot {x'_{\text{RGB}}}^t \tag{2}$$

$$h_{\text{Event}}^t = \overline{A}_{\text{Event}} \cdot h_{\text{Event}}^{t-1} + \underbrace{\mathcal{M}(\overline{B}_{\text{Event}}, \overline{B}_{\text{RGB}})}_{\text{Joint Optimization}} \cdot {x'_{\text{Event}}}^t \tag{3}$$

$$y_{\text{Event}}^t = \underbrace{\mathcal{M}(C_{\text{Event}}, C_{\text{RGB}})}_{\text{Joint Optimization}} \cdot h_{\text{Event}}^t + D_{\text{Event}} \cdot {x'_{\text{Event}}}^t \tag{4}$$

Multi-modal joint optimization function $\mathcal{M}$ is defined as:

$$\mathcal{M}(\mathcal{A}, \mathcal{B}) = \left( W \cdot [\mathcal{A}; \mathcal{B}] + b \right) \oplus \mathcal{A} \tag{5}$$

where $[;]$ denotes concatenation and $\oplus$ represents element-wise addition. This formulation enables feature concatenation from both modalities followed by learnable projections ($W, b$), creating shared parameters that preserve modality-specific characteristics while establishing cross modality associations.

### 3.1.2 MCIB.

We further introduce the Multi-modal Collaborative Interaction Block (MCIB) to achieve fine-grained semantic alignment and complementary fusion between modalities. As depicted in Fig. 2, MCIB processes input features $\mathbf{y}_{\text{RGB}}$ and $\mathbf{y}_{\text{Event}}$ through a cross-attention and projects them into each other's Query. This enables RGB features to actively retrieve dynamic details from event streams while event features selectively attend to static contextual semantics within RGB data. Following, we take the projection of two modalities as each other's Query:

$$Q_{\text{RGB}} = W_Q^{\text{RGB}} \mathbf{y}_{\text{Event}}, \; Q_{\text{Event}} = W_Q^{\text{Event}} \mathbf{y}_{\text{RGB}} \tag{6}$$

$$\mathbf{y}'_{\text{RGB}} = \text{MHA}(\mathbf{y}_{\text{RGB}}), \mathbf{y}'_{\text{Event}} = \text{MHA}(\mathbf{y}_{\text{Event}}) \tag{7}$$

where $\mathbf{y_{RGB}}$ and $\mathbf{y_{Event}}$ are input features of semantic fusion block, $W$ is learnable linear projection. To adaptively balance modality contributions, we design a gating mechanism:

$$\alpha = \sigma\left(\sum \left(\mathbf{y_{RGB}} \odot \mathbf{y_{Event}}\right)\right) \tag{8}$$

$$H = \alpha \cdot \mathbf{y}'_{\mathbf{RGB}} + (1 - \alpha) \cdot \mathbf{y}'_{\mathbf{Event}} \tag{9}$$

where $\sigma$ denotes the sigmoid function and $\odot$ represents element-wise multiplication. This gating strategy maintains inter-modality equilibrium without introducing additional parameters and avoid biasing towards a certain mode, effectively reducing model complexity while preserving fusion flexibility.

### 3.2 HCE-MOE

As described in Intro. 1, we try to adopt Mixture-of-Experts (MoE) to dynamically select experts to obtain diverse semantic representations. Unlike conventional MoE, we design Heterogeneous Experts to minimize semantic overlap and maximize diversity. To address the limitations of linear routing in standard MoE (Mustafa et al., 2022), load imbalance, and poor global context modeling, we introduce an Attention-Guided Router inspired by Wu et al. (2024b), dynamically adapting to complex feature relationships. So, we propose the HCE-MoE, which contains: **Router with Attention**, **Deep Expert**, **Attention Expert**, and **Focal Expert.**

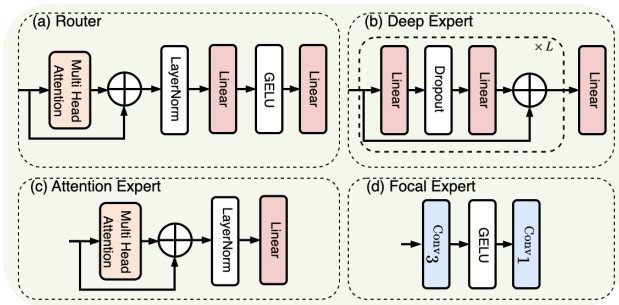

Figure 3: Structure of the proposed HCE-MoE.

### 3.2.1 ROUTER WITH ATTENTION.

Different from attention router (Wu et al., 2024b; Blecher & Fine, 2023b), we further combine the original features and attention-processed features and add nonlinear representations to improve router capabilities.

$$H' = \mathcal{P}_2\left(\sigma\left(\mathcal{P}_1(\mathcal{LN}(\text{MHA}(H) \oplus H))\right)\right) \tag{10}$$

$$\mathcal{G} = \{(m_i, h_i)\}_{i=1}^k = \text{TopK}\left(\text{Softmax}(H')\right) \tag{11}$$

where $H' \in \mathbb{R}^{N_e}$, $N_e$ is number of experts. $\mathcal{P}_{1/2}$ are linear projections, $\text{MHA}(\cdot)$ denotes multi-head attention, and $\mathcal{G}$ contains selected expert indices $m_i$ with corresponding weights $h_i$, Top-K takes the first $k$ maximum value operation. Compared with traditional MoE routers (Shazeer et al., 2017; Zhou et al., 2022a), our structure can more carefully model the mapping relationship input to experts through hierarchical processing of $\mathcal{P}_1 \to \sigma \to \mathcal{P}_2$, reducing routing confusion. Compared with existing methods of using attention routers (Blecher & Fine, 2023a; Wu et al., 2024a), we combine the output of MHA and the original input, the router can simultaneously utilize the original features without attention processing and the global context of attention output, avoiding information loss while fusing context information.

### 3.2.2 DEEP EXPERT.

Traditional expert (Zhou et al., 2022b; Wang et al., 2024a) typically employ 1 or 2 layers, shallow networks strive to compose low-level features into high-level semantics, while narrow intermediate layers restrict the model's ability to handle complex features. So, our deep expert implements progressive feature refinement through $L$ stacked transformation blocks, each of the $L$ layers expands the input dimension to $4\times$ its original size, creating a high-dimensional space for features. Subsequent compression back to the original dimension ensures compatibility with residual connections while preserving key information:

$$O_D = \mathcal{P}_3\left(\left(\bigcirc_{l=1}^{L}\left[\mathcal{P}_2^{(l)} \circ \mathcal{D}^{(l)} \circ \mathcal{P}_1^{(l)}\right]\right)(H)\right) \tag{12}$$

where $H$ is input feature, $\mathcal{P}_1^{(l)} : \mathbb{R}^d \to \mathbb{R}^{4d}$ expands dimensions, $\mathcal{P}_2^{(l)} : \mathbb{R}^{4d} \to \mathbb{R}^d$ restore dimensions, $\mathcal{P}_3^{(l)} : \mathbb{R}^d \to \mathbb{R}^{\mathcal{J}}$, $\mathcal{J}$ is number of expression class, $\mathcal{D}$ denotes dropout, $l$ is $l-th$ layer, $L$ is depth and $\bigcirc_{l=1}^{L}$ is nested operations from the first layer to the $L$ layer.

### 3.2.3 ATTENTION EXPERT.

Ordinary experts (Zhou et al., 2022b; Wang et al., 2024a) are usually composed of only fully connected layers, lack explicit sequence modeling capabilities, and are difficult to deal with scenarios that require global context. While deep-oriented architectures excel at hierarchical feature extraction, they lack input-adaptive feature weighting crucial for emotion-varying contexts. To address this, we design attention experts specializing in global contextual reasoning through multi-head self-attention. Each expert introduces MHA internally to enable it to capture dependencies in the input sequence, which is crucial for sequence processing tasks:

$$O_A = \mathcal{P}(\mathcal{LN}(\text{MHA}(H) \oplus H)) \tag{13}$$

| Methods | | Metrics(%) | | Accuracy under lighting conditions(%) | | | | Accuracy of emotion classification(%) | | | | | | |
|---|---|---|---|---|---|---|---|---|---|---|---|---|---|---|
| | | WAR | UAR | Normal | Overexposure | Low-Light | HDR | Happy | Sadness | Anger | Disgust | Surprise | Fear | Neutral |
| Former DFER | Face | 65.8 | 67.2 | 70.1 | 65.4 | 66.2 | 61.1 | 81.5 | 75.2 | 85.8 | 59.4 | 39.3 | 50.8 | 78.6 |
| Former DFER* | Face | 48.0 | 48.0 | 47.0 | 51.9 | 45.6 | 47.2 | 44.1 | 65.2 | 46.0 | 66.5 | 28.0 | 50.3 | 36.1 |
| R(2+1)D | Face | 49.7 | 51.5 | 54.3 | 50.3 | 44.4 | 49.3 | 63.6 | 45.5 | 65.7 | 27.8 | 33.3 | 37.9 | 86.6 |
| 3D Resnet18 | Face | 49.1 | 50.5 | 51.9 | 51.4 | 44.8 | 47.8 | 54.8 | 45.4 | 67.7 | 23.8 | 37.2 | 42.8 | 81.6 |
| Resnet50 + GRU | Face | 35.2 | 34.7 | 43.0 | 35.7 | 28.9 | 32.8 | 27.9 | 38.0 | 49.7 | 44.5 | 6.9 | 70.0 | 5.6 |
| Resnet18 + LSTM | Face | 56.3 | 58.0 | 57.9 | 60.4 | 53.9 | 52.5 | 57.8 | 86.0 | 64.9 | 46.5 | 9.2 | 81.6 | 59.8 |
| EMO | Eye | 63.1 | 63.3 | 61.8 | 62.8 | 60.1 | 69.6 | 75.0 | 75.1 | 70.2 | 48.1 | 37.5 | 54.1 | 82.8 |
| EMO* | Eye | 53.2 | 53.3 | 46.1 | 60.2 | 55.5 | 58.9 | 62.0 | 73.2 | 60.1 | 38.7 | 25.7 | 48.0 | 65.3 |
| Eyemotion | Eye | 78.8 | 79.5 | 79.0 | 81.8 | 81.5 | 72.5 | 74.3 | 85.5 | 79.5 | 74.3 | 69.1 | 79.2 | 94.5 |
| Eyemotion* | Eye | 75.9 | 77.2 | 77.8 | 75.9 | 79.8 | 69.7 | 79.6 | 85.7 | 81.2 | 71.2 | 54.7 | 71.6 | **96.4** |
| SEEN | Eye | 83.6 | 84.1 | 83.3 | 85.6 | 80.8 | 84.8 | 85.0 | 89.9 | 92.2 | 76.7 | 72.1 | 87.7 | 85.2 |
| MSKD | Eye | 86.2 | 86.6 | 84.4 | 89.1 | 88.3 | 82.7 | 85.6 | 91.7 | 92.3 | 79.0 | 79.4 | 88.0 | 90.3 |
| HI-Net | Eye | 86.9 | 87.7 | 84.6 | 90.3 | 87.2 | 85.2 | 93.4 | 95.5 | 87.8 | 85.3 | 70.6 | **91.2** | 89.8 |
| **Ours** | Eye | **91.3** | **91.9** | **87.7** | **93.8** | **93.6** | **90.2** | **97.0** | **98.4** | **94.7** | **87.7** | **81.7** | 89.4 | 94.6 |

Table 1: Comparison with SOTA methods on SSE dataset under UAR, WAR, Normal, Overexposure, Low-Light and HDR. Best results are shown in **bold**, the second best results are shown in underlined. * indicates that the model has not been pre-trained.

where $O_A \in \mathbb{R}^{\mathcal{J}}$, $\mathcal{J}$ is number of expression class, $\mathcal{LN}$ is Layer Normalization and $\mathcal{P}$ is linear projection. Add the output of MHA with the original input, retaining the underlying feature information to avoid information loss. In addition, layer normalization is used after MHA and residual connection to improve overfitting and enhance model generalization.

### 3.2.4 FOCAL EXPERT.

The efficacy of Single-eye expression recognition heavily relies on the precise localization of micro-expression patterns (e.g., eye narrowing or brow furrowing). While global attention mechanisms excel at contextual modeling, they often overlook fine-grained spatial details critical for subtle emotion discrimination. To achieve this, we design Focal expert specializing in hierarchical local features:

$$O_C = C_1(\sigma(C_3(H))) \tag{14}$$

where $O_C \in \mathbb{R}^{\mathcal{J}}$, $\mathcal{J}$ is number of expression class, $C_1$ and $C_3$ denotes convolution with kernel size 1 and 3.

### 3.2.5 EXPERT INTEGRATION.

Final prediction integrate outputs from activated experts through weighted aggregation:

$$O = \sum_{(m_i, h_i) \in \mathcal{G}} h_i \cdot O_{m_i}(H) \tag{15}$$

where $(m_i, h_i) \in \mathcal{G}$ calculated from router with attention, $\sum h_i = 1$ ensures normalized contributions, $O_{m_i}(\cdot)$ is the activated expert, maybe $O_D$, $O_A$, $O_C$.

### 3.3 LOSS FUNCTION

To prevent routers from overly favoring a specific combination of experts and encourage expert diversity, we add constraints to HCE-MoE:

$$\mathcal{L}_{Ortho} = MSE(W^\mathsf{T} W, I) \tag{16}$$

where $MSE$ is the mean square error, $W$ is the weight of the last layer (output layer) of the router and $I$ is the identity matrix. Overall, loss of MCO-E Net is the cross entropy loss and $\mathcal{L}_{Ortho}$, as follows:

$$\mathcal{L} = \mathcal{L}_{CE} + \mathcal{L}_{Ortho} \tag{17}$$

We verify $\mathcal{L}_{Ortho}$ is effective in Appendix A.4.

## 4 EXPERIMENT

**Experimental details. Datasets:** Our method is extensively experimented on the SEE (Zhang et al., 2023a) and DSEE (Wang et al., 2024b) datasets. **Evaluation Metrics:** Performance is measured via:

UAR (Unweighted Average Recall) and WAR (Weighted Average Recall). In addition, the efficiency analysis is given in the Appendix A.1, more ablation studies are given in the Appendix A.3 , more details on hyperparameter settings are given in the Appendix A.6.

## 4.1 Comparison with State-of-the-art Methods

Since combining RGB and Event in Single-eye expression recognition task is a new strategy, related work is scarce. Therefore, to prove the performance of our network, our network is compared with RGB-based methods: EMO (Wu et al., 2020b), Eyemotion (Hickson et al., 2017), Former DFER (Zhao & Liu, 2021), R(2+1)D (Tran et al., 2018), 3D Resnet18 (Hara et al., 2018), Resnet50+GRU (Cho, 2014), Resnet18+ LSTM (Hochreiter & Schmidhuber, 1997) and Event-based methods: SEEN (Zhang et al., 2023a), MSKD (Wang et al., 2024b) and HI-Net (Han et al., 2025). In addition, expression recognition methods categorize into eye-based or face-based analysis, as summarized in Table 1.

As shown in Table 1 and Table 3, on the SEE (Zhang et al., 2023a) and DSEE (Wang et al., 2024b) datasets, our proposed method significantly outperforms the existing SOTA methods in WAR and UAR. Specifically, on SEE, our method outperforms the SOTA methods by $4.4\%$ and $4.2\%$, respectively. On DSEE, our method outperforms the SOTA methods by $2.0\%$ and $1.8\%$, respectively. In addition, we also show the performance under four different lighting conditions. As shown in Table 1, our method shows the best accuracy under all lighting conditions. In addition, compared with SEEN (Zhang et al., 2023a) and HI-Net (Han et al., 2025), our method shows better ability to capture and emphasize semantic features around the eye or eyebrow region under four lighting conditions, as shown in Fig. 4.

## 4.2 Ablation study

| Methods | WAR | UAR |
|---|---|---|
| A. RGB Only | 83.5 | 84.3 |
| B. Event Only | 72.6 | 73.3 |
| C. w/o MCO-Mamba | 89.4 | 90.1 |
| D. MCO-Mamba(w/o MJOS) | 89.9 | 90.5 |
| E. MCO-Mamba(w/o MCIB) | 89.8 | 90.5 |
| F. w/o HCE-MoE | 88.9 | 89.7 |
| G. HCE-MoE (1 type of expert) | 89.9 | 90.5 |
| H. HCE-MoE (2 type of experts) | 90.6 | 91.3 |
| I. Ours | 91.3 | 91.9 |

Table 2: Results produced by combining different components of our proposed network.

### 4.2.1 Effectiveness of Multi-modality Semantics.

Table. 2 demonstrates the critical advantage of multimodal fusion through controlled experiments: Exp. A (RGB only) achieves 83.5% WAR while Exp. B (Event only) reaches 72.6% WAR, whereas our full RGB+Event model attains 91.3% WAR. This significant gain stems from complementary strengths: event streams capture high-temporal-resolution dynamic details with extreme lighting robustness, while RGB provides rich spatial-semantic around ocular regions, enabling comprehensive expression modeling.

### 4.2.2 Effectiveness of MCO-Mamba.

In proposed MCO-Mamba, we use MJOS (for Mamba) and MCIB block for joint optimization of model and Multi-modal collaborative modeling, respectively. From the experimental results shown in $C$ to $E$ of Table. 2, we can analyze the greatest impact on performance is row $C$. To the removal of MCO-Mamba, the advantageous semantics of these two modalities cannot be combined, resulting in performance degradation. $D$ of Table. 2 removes MJOS from MCO-Mamba. This means that the features extracted by backbone are directly fusion through MCIB block. The lack of parameter joint optimization strategy leads to significant reduction in model performance. $E$ of Table. 2 is remove MCIB from MCO-Mamba, change the fusion method to feature addition. The lack of semantic integration leads to performance degradation in WAR and UAR.

| Methods | | Metrics(%) | | Accuracy under lighting conditions(%) | | | | Accuracy of emotion classification(%) | | | | | | |
|---|---|---|---|---|---|---|---|---|---|---|---|---|---|---|
| | | WAR | UAR | Normal | Overexposure | Low-Light | HDR | Happy | Sadness | Anger | Disgust | Surprise | Fear | Neutral |
| Former DFER | Face | 59.7 | 61.1 | 58.1 | 64.2 | 60.7 | 58.8 | 71.1 | 54.8 | 67.2 | 64.3 | 45.2 | 42.7 | 82.6 |
| Former DFER* | Face | 50.2 | 51.1 | 50.3 | 50.7 | 47.3 | 53.5 | 63.1 | 50.7 | 49.9 | 48.1 | 40.8 | 42.0 | 62.8 |
| R(2+1)D | Face | 45.8 | 45.0 | 49.1 | 45.9 | 36.6 | 44.4 | 52.1 | 53.5 | 27.1 | 62.0 | 54.7 | 29.8 | 35.6 |
| 3D Resnet18 | Face | 53.3 | 53.8 | 54.9 | 55.6 | 44.2 | 56.5 | 62.7 | 56.5 | 58.9 | 50.9 | 51.5 | 33.5 | 62.6 |
| Resnet50 + GRU | Face | 71.6 | 66.3 | 69.9 | 70.5 | 69.3 | 78.4 | 66.3 | 62.2 | 78.3 | 75.8 | 68.5 | 65.7 | 84.7 |
| Resnet18 + LSTM | Face | 72.2 | 73.0 | 71.3 | 73.1 | 69.7 | 79.6 | 72.3 | 61.9 | 78.3 | 76.8 | 69.9 | 66.9 | 84.8 |
| EMO | Eye | 68.0 | 68.8 | 67.0 | 68.2 | 65.5 | 76.5 | 73.7 | 59.2 | 70.8 | 74.2 | 61.6 | 61.7 | 80.2 |
| EMO* | Eye | 67.8 | 68.7 | 67.4 | 67.0 | 63.2 | 78.6 | 68.3 | 64.4 | 70.7 | 73.7 | 62.9 | 58.6 | 82.5 |
| Eyemotion | Eye | 72.3 | 73.1 | 71.3 | 72.1 | 69.3 | 82.0 | 72.3 | 66.4 | 76.9 | 74.9 | 70.4 | 64.9 | 85.8 |
| Eyemotion* | Eye | 71.8 | 72.7 | 70.4 | 70.4 | 70.1 | 83.8 | 71.0 | 68.2 | 75.0 | 74.8 | 66.8 | 67.2 | 86.3 |
| SEEN | Eye | 71.9 | 72.6 | 70.9 | 74.8 | 69.8 | 75.5 | 72.2 | 65.3 | 78.5 | 74.9 | 72.0 | 61.0 | 84.2 |
| MSKD | Eye | 77.4 | 77.9 | 76.1 | 78.6 | 75.0 | 86.2 | 80.1 | 73.7 | 83.8 | 79.3 | 75.4 | 66.8 | 86.3 |
| HI-Net | Eye | 72.4 | 73.3 | 71.7 | 72.5 | 66.6 | 86.8 | 78.0 | 68.9 | 68.8 | 74.3 | 76.6 | 58.9 | 87.5 |
| **Ours** | Eye | **79.4** | **79.7** | **79.1** | 75.2 | **75.1** | **96.5** | **85.5** | 69.7 | 81.2 | **82.2** | **82.5** | **70.9** | 85.9 |

Table 3: Comparison with SOTA methods on DSSE dataset under UAR, WAR, Normal, Overexposure, Low-Light and HDR. Best results are shown in **bold**, the second best results are shown in underlined. * indicates that the model has not been pre-trained.

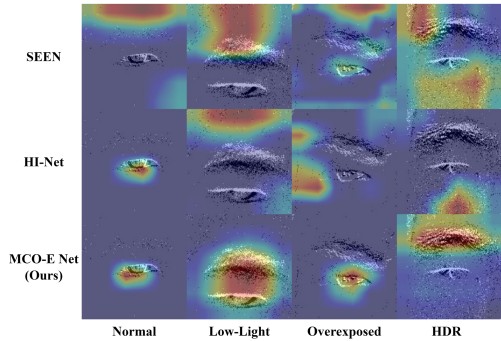

Figure 4: Heatmap visualization of the comparison between our proposed MCO-E Net and SOTA methods

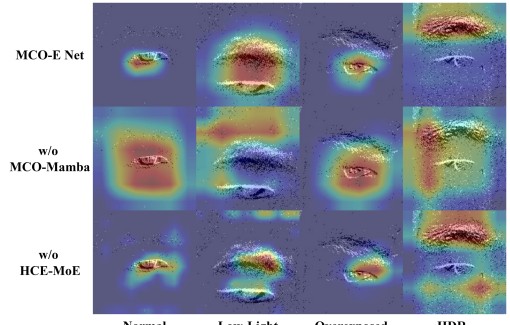

Figure 5: Heatmap visualization of the comparison between our proposed MCO-E Net and removing MCO-Mamba or HCE-MoE

### 4.2.3 EFFECTIVENESS OF HCE-MoE.

Ablation studies for HCE-MoE (Table 2) confirm its critical role: Replacing HCE-MoE with a fully-connected layer reduced WAR/UAR by 2.4%/2.2%, demonstrating that HCE-MoE decoupling is essential for capturing fine-grained patterns (e.g., edge/texture features). Simplified variants using one expert type ($G$) or two types ($H$) consistently underperformed full HCE-MoE, proving heterogeneous expert diversity (Deep/Attention/Focal Experts) reduces feature oversight probability. Visual analysis (Fig. 5) further shows degraded focus in lighting variations without HCE-MoE or MCO-Mamba.

For more ablation studies on MCO-Mamba and HCE-MoE, we have included them in the Appendix A.3.

## 5 CONCLUSION

In this paper, we proposed a Multi-modal Collaborative Optimization and Expansion Network (MCO-E Net), for the single-eye expression recognition tasks. The MCO-E Net contains two novel designs: Multi-modal Collaborative Optimization Mamba (MCO-Mamba), Heterogeneous Collaborative and Expansion MoE (HCE-MoE). The MCO-Mamba drove the model to balance the learning of two-modal semantics and capture the advantages of both modalities through joint optimization in Mamba. The HCE-MoE systematically combines multiple feature expertise through a heterogeneous architecture, enabling collaborative learning of complementary semantics and capturing comprehensive semantics. Extensive experiments demonstrate that our MCO-E Net achieves competitive performance on the single-eye expression recognition task.

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

## A   APPENDIX

Anonymous link to our code repository: `https://anonymous.4open.science/r/MCO-E-Net-459B`

### A.1   EFFICIENCY ANALYSIS.

As shown in Table. 4, our method achieves 91.3% WAR, while attaining the fastest inference speed. This accuracy-speed synergy demonstrates strong potential for real-time applications.

| Methods | WAR | FLOPs (G) | Time (ms) |
|---|---|---|---|
| Eyemotion (Hickson et al., 2017) | 78.8 | 5.73 | 17.5 |
| EMO (Wu et al., 2020b) | 63.1 | 0.32 | 7.1 |
| SEEN (Zhang et al., 2023a) | 83.6 | 0.95 | 7.2 |
| MSKD (Wang et al., 2024b) | 86.2 | **0.27** | 6.1 |
| HI-Net (Han et al., 2025) | 86.9 | 11.27 | 7.4 |
| Ours | **91.3** | 1.77 | **2.3** |

Table 4: Computational efficiency comparison of eye-based expression recognition methods.

### A.2   ALGORITHM OF MJOS FOR MAMBA

---

**Algorithm 1** Process of Mamba with MJOS

---

**Input:** sequence $\mathbf{E}, \mathbf{F}$
**Output:** sequence $\mathbf{y_{RGB}}, \mathbf{y_{Event}}$
1: $\mathbf{X_{RGB}}, \mathbf{Z_{RGB}} \leftarrow \mathbf{Lin^{X_{RGB}}}(\mathbf{F}), \mathbf{Lin^{Z_{RGB}}}(\mathbf{F})$
2: $\mathbf{X_{Event}}, \mathbf{Z_{Event}} \leftarrow \mathbf{Lin^{X_{Event}}}(\mathbf{E}), \mathbf{Lin^{Z_{Event}}}(\mathbf{E})$
3: $\mathbf{X'_{RGB}}, \mathbf{X'_{Event}} \leftarrow \mathbf{Conv1d}(\mathbf{X_{RGB}}), \mathbf{Conv1d}(\mathbf{X_{Event}})$
4: **for** $o$ in $\{$forward,backward$\}$ **do**
5:   /* Parameter initialization */
6:   $\mathbf{A^o_{RGB}}, \mathbf{A^o_{Event}} \leftarrow$ Parameter initialization
7:   $\mathbf{B^o_{RGB}}, \mathbf{B^o_{Event}} \leftarrow \mathbf{Lin^{B^o_{RGB}}}(\mathbf{X'_{RGB}}), \mathbf{Lin^{B^o_{Event}}}(\mathbf{X'_{Event}})$
8:   $\mathbf{B^o_{Fusion}} \leftarrow \mathbf{Concat}(\mathbf{B^o_{RGB}}, \mathbf{B^o_{Event}})$
9:   $\mathbf{B^{o'}_{RGB}} \leftarrow \mathbf{Lin^{B^{o'}_{RGB}}}(\mathbf{B^o_{Fusion}}) + \mathbf{B^o_{RGB}}$
10:   $\mathbf{B^{o'}_{Event}} \leftarrow \mathbf{Lin^{B^{o'}_{Event}}}(\mathbf{B^o_{Fusion}}) + \mathbf{B^o_{Event}}$
11:   $\mathbf{C^o_{RGB}}, \mathbf{C^o_{Event}} \leftarrow \mathbf{Lin^{C^o_{RGB}}}(\mathbf{X'_{RGB}}), \mathbf{Lin^{C^o_{Event}}}(\mathbf{X'_{Event}})$
12:   $\mathbf{C^o_{Fusion}} \leftarrow \mathbf{Concat}(\mathbf{C^o_{RGB}}, \mathbf{C^o_{Event}})$
13:   $\mathbf{C^{o'}_{RGB}} \leftarrow \mathbf{Lin^{C^{o'}_{RGB}}}(\mathbf{C^o_{Fusion}}) + \mathbf{C^o_{RGB}}$
14:   $\mathbf{C^{o'}_{Event}} \leftarrow \mathbf{Lin^{C^{o'}_{Event}}}(\mathbf{C^o_{Fusion}}) + \mathbf{C^o_{Event}}$
15:   $\mathbf{D^o_{RGB}}, \mathbf{D^o_{Event}} \leftarrow 1$
16:   /* Discretize */
17:   $\mathbf{\Delta^o_{RGB}} \leftarrow \log(1+ \exp(\mathbf{Lin^{\Delta^o_{RGB}}}(\mathbf{X'_{RGB}})+\mathbf{Param^{\Delta^o_{RGB}}}))$
18:   $\mathbf{\Delta^o_{Event}} \leftarrow \log(1+ \exp(\mathbf{Lin^{\Delta^o_{Event}}}(\mathbf{X'_{Event}})+\mathbf{Param^{\Delta^o_{Event}}}))$
19:   $\mathbf{\overline{A}^o_{RGB}}, \mathbf{\overline{B}^{o'}_{RGB}} \leftarrow \mathbf{discretize}(\mathbf{\Delta^o_{RGB}}, \mathbf{A^o_{RGB}}, \mathbf{B^{o'}_{RGB}})$
20:   $\mathbf{\overline{A}^o_{Event}}, \mathbf{\overline{B}^{o'}_{Event}} \leftarrow \mathbf{discretize}(\mathbf{\Delta^o_{Event}}, \mathbf{A^o_{Event}}, \mathbf{B^{o'}_{Event}})$
21:   /* State Space Model */
22:   $\mathbf{y^o_{RGB}} \leftarrow \mathbf{SSM}(\mathbf{\overline{A}^o_{RGB}}, \mathbf{\overline{B}^{o'}_{RGB}}, \mathbf{C^{o'}_{RGB}}, \mathbf{D^o_{RGB}})(\mathbf{x'_{RGB}})$
23:   $\mathbf{y^o_{Event}} \leftarrow \mathbf{SSM}(\mathbf{\overline{A}^o_{Event}}, \mathbf{\overline{B}^{o'}_{Event}}, \mathbf{C^{o'}_{Event}}, \mathbf{D^o_{Event}})(\mathbf{x'_{Event}})$
24: **end for**
25: $\mathbf{y_{RGB}} \leftarrow \mathbf{Lin^{y_{RGB}}}(\mathbf{Z_{RGB}} \cdot (\mathbf{y^{forward}_{RGB}} + \mathbf{y^{backward}_{RGB}}))$
26: $\mathbf{y_{Event}} \leftarrow \mathbf{Lin^{y_{Event}}}(\mathbf{Z_{Event}} \cdot (\mathbf{y^{forward}_{Event}} + \mathbf{y^{backward}_{Event}}))$
27: **return** $\mathbf{y_{RGB}}, \mathbf{y_{Event}}$

---

Table 5: Results produced by change the number of experts in HCE-MoE

| Methods | WAR | UAR |
|---|---|---|
| A. $N_e = 5$ | 90.0 | 90.6 |
| B. $N_e = 10$ | 89.0 | 89.7 |
| C. $N_e = 15$ | 89.5 | 90.3 |
| D. Ours ($N_e = 8$) | 91.3 | 91.9 |

## A.3 MORE ABLATION STUDIES

### A.3.1 NUMBER OF EXPERTS IN HCE-MOE.

Table. 5 demonstrates the impact of varying the number of experts ($N_e$) in our HCE-MoE architecture on WAR and UAR. While configuration $A$ ($N_e = 5$) achieves metrics of 90.0% WAR and 90.6% UAR, increasing the expert count to 10 (configuration $B$) paradoxically degrades performance to 89.0% WAR and 89.7% UAR. Subsequent expansion to 15 experts (configuration $C$) yields partial recovery (89.5% WAR, 90.3% UAR), yet still underperforms relative to the baseline. The result demonstrating that intermediate expert counts enable more effective knowledge specialization. This optimal balance suggests that: Insufficient experts limit model capacity for capturing complex pattern variations; Excessive experts introduce parameter complexities, resulting in reduced performance.

Table 6: Results produced by change the $Topk$ of HCE-MoE

| Methods | WAR | UAR |
|---|---|---|
| A. $k = 1$ | 89.6 | 90.3 |
| B. $k = 3$ | 90.3 | 90.9 |
| C. $k = 5$ | 90.4 | 91.0 |
| D. Ours ($k = 2$) | 91.3 | 91.9 |

**Topk of HCE-MoE.** The experimental results in Table. 6 demonstrate a key balance in expert activation for our HCE-MoE framework. While extending the activation experts from $k = 1$ to $k = 3$ improves recognition accuracy, further upgrading to $k = 5$ yields only marginal gains, suggesting the inherent limitations of indiscriminate expert aggregation. Our configuration with $k = 2$ achieves optimal performance, outperforming both under activated and over activated settings by significant margins.

### A.3.2 DISCUSSION OF MCO-MAMBA

Table 7: Results produced by different SSM matrix Interaction of our proposed MCO-Mamba.

| Methods | WAR | UAR |
|---|---|---|
| A. MCO-Mamba ($A$) | 90.2 | 90.8 |
| B. MCO-Mamba ($B$) | 90.2 | 90.8 |
| C. MCO-Mamba ($C$) | 90.1 | 90.7 |
| D. MCO-Mamba ($D$) | 89.8 | 90.5 |
| E. MCO-Mamba ($A, B$) | 89.9 | 90.6 |
| F. MCO-Mamba ($A, C$) | 90.2 | 90.9 |
| G. MCO-Mamba ($A, D$) | 89.5 | 90.2 |
| H. MCO-Mamba ($B, D$) | 89.8 | 90.4 |
| I. MCO-Mamba ($C, D$) | 90.1 | 90.7 |
| J. MCO-Mamba ($A, B, C, D$) | 90.0 | 90.6 |
| K. MCO-Mamba ($B, C$) | 91.3 | 91.9 |

To verify the effectiveness of MCO-Mamba, we designed experiments from three perspectives: parameter sharing, selection of interactive parameters, and exchange of parameters.

**Parameter sharing.** As mentioned in related work, static interaction strategies (such as simple parameter sharing or exchanging) that force a unified feature space tend to weaken modality-specific features and cause an imbalance between modality-shared and modality-specific features. To demonstrate that the static strategy that only sharing parameters between modalities affects modality-specific representations, we designed experiments $A$ to $D$ of Table. 7. $A$ to $D$ respectively indicate that the Multi-modal Joint Optimization Scheme (MJOS) is removed in our MCO-Mamba, and only a single parameter matrix $A$, $B$, $C$ or $D$ is shared. From the results, we can see that the performance drops dramatically. This is because the shared SSM parameter matrix affects the modality-specific representation, resulting in an imbalance in the representation between modalities.

**Selection of interactive parameters.** To validate our choice of interacting $B$ and $C$, we designed experiments $E$ to $I$ of Table. 7, which represent interactive operations $\mathcal{M}$ on different parameter matrices. From the analysis $E$ to $I$ of Table. 7, we can see that the performance has dropped significantly due to the $A$ and $D$. $A$ interacts between two modalities, resulting in unstable state transitions. $D$ is a residual term, and interacting with $D$ will affect the introduction of original information.

**Exchange of parameters.** As shown in row $J$ of Table. 7, we exchange all SSM parameters of the two modalities. The observed performance degradation stems from the oversimplified exchange mechanism undermining the critical modal-sharing features, which are essential for maintaining synergistic coupling between multi-modality.

### A.3.3 DISCUSSION OF HCE-MOE

Table 8: Results produced by change the router

| Methods | WAR | UAR |
|---|---|---|
| $A$. MLP | 88.9 | 89.7 |
| $B$. Deep experts Only. | 90.2 | 90.9 |
| $C$. Attention Experts Only | 89.9 | 90.6 |
| $D$. Focal Experts Only | 89.9 | 90.5 |
| $E$. Ours | 91.3 | 91.9 |

The ablation study on router architectures as shown in Table. 8, where we designed four experiment $A$ to $D$: $A$ is to exchange HCE-MoE for MLP; $B$ is to keep only Deep expert in HCE-MoE; $C$ is to keep only Attention Experts in HCE-MoE; $B$ is to keep only Focal Experts in HCE-MoE. our proposed routing mechanism $E$ achieves state-of-the-art performance. The MLP baseline ($A$) produces the weakest results, and our designed expert show clear advantages: the Deep Expert ($B$) achieves 90.2% WAR, slightly better than the Attention Expert ($C$) and Focal Expert ($D$), indicating that the deep expert provides slightly greater discriminative power for this task.

### A.4 ORTHOGONAL LOSS

In order to verify the effectiveness of the Orthogonal Loss we used in HCE-MoE, we recorded the proportion of the expert's activation and the proportion of the weight in the experiment.

Fig. 6 demonstrates a remarkably consistent equilibrium in our Heterogeneous Collaborative and Expansion Mixture-of-Experts (HCE-MoE). In the activation landscape (left), all three expert types maintain near-identical utilization rates across 200 training epochs, with trajectories strictly bound within 5% deviation from the theoretical equipartition line (30%). The inter-expert fluctuation range remains confined below 2% at any sampled epoch (e.g., epoch 100: Focal 31.2%, Attention 29.8%, Deep 30.1%).

This equilibrium extends to parameter allocation, where the weight distribution (right) forms invariant proportional bands: Focal (blue, baseline), Attention (red, middle), and Deep (green, upper) experts perpetually occupy fixed 33.3%0.5% partitions of the total parametric resources. The perfect superimposition of weight boundaries at all epochs confirms an explicit anti-collapse mechanism, as neither specialist dominates nor diminishes in representational capacity.

As evidenced by the t-SNE projection in Fig. 7, the distinct spatial segregation among features, represented by blue circles (Focal), green crosses (Attention), and red diamonds (Deep), demonstrates

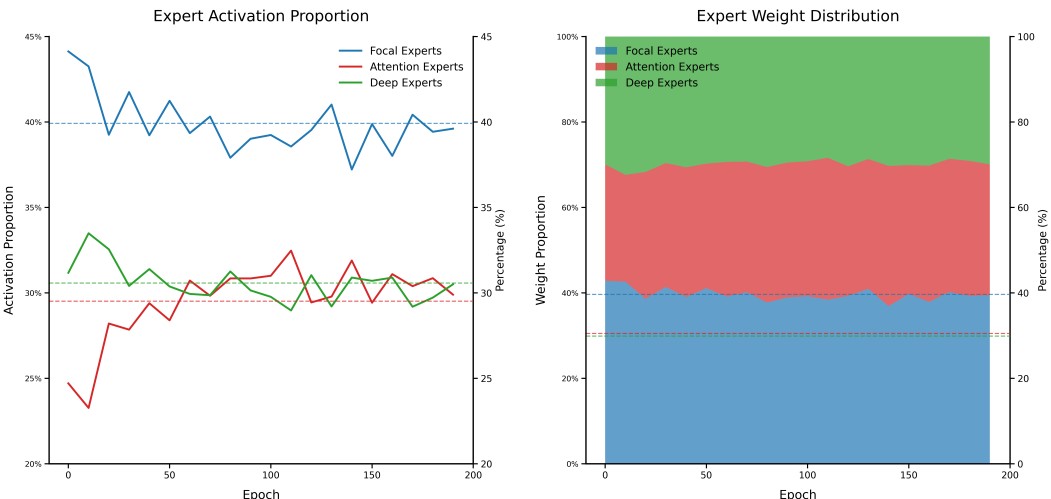

Figure 6: Activation percentage of different experts during training.

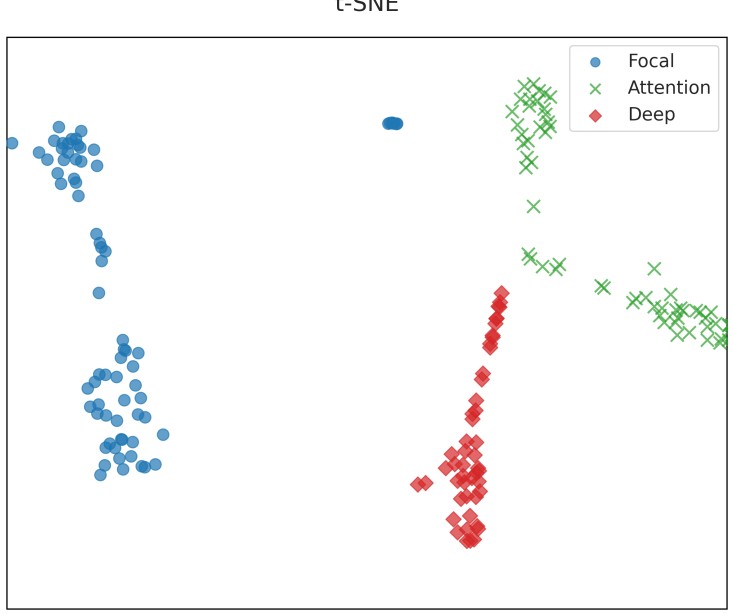

Figure 7: t-SNE of features extracted by different experts.

fundamental divergence in representational space across expert types. The Focal experts concentrate within a dense low-variance cluster. In contrast, Attention experts exhibit deliberate dispersion across the central band. The Deep experts vertically stratify along high-dimensional boundaries.

Spatial isolation of clusters confirms orthogonal knowledge extraction, where collectively enabling complementary representation learning without collision.

### A.5 VISUAL COMPARISON OF MCO-E NET AND OTHER SOTA METHODS

As shown in Fig. 8, our method demonstrates superior capability in capturing and emphasizing semantic features around eye or eyebrow regions across four lighting conditions when compared with SEEN (Zhang et al., 2023a) and HI-Net (Han et al., 2025). This focused perception enhances expression classification accuracy.

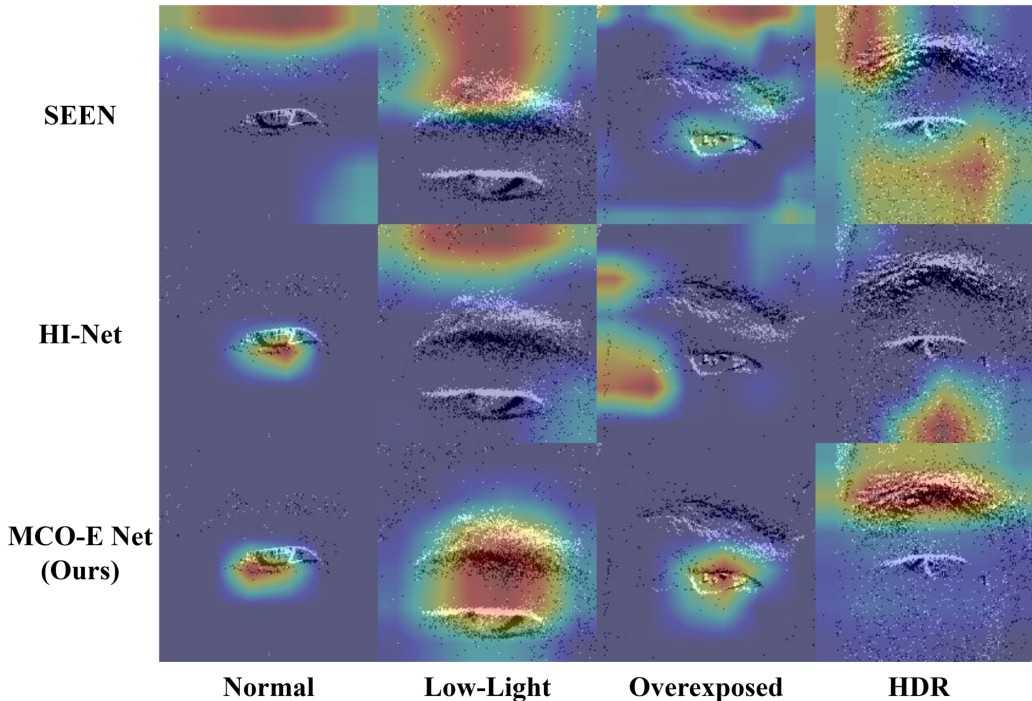

SEEN

HI-Net

MCO-E Net
(Ours)

| Normal | Low-Light | Overexposed | HDR |

Figure 8: Heatmap visualization of the comparison between our proposed MCO-E Net and SOTA methods

Overall, experimental results confirm that our proposed network performance breakthrough ($91.3\%$ and $91.9\%$ in WAR and UAR) is attributed to two core innovations: MCO-Mamba enabling dynamic cross modality parameter interaction and semantic fusion, and HCE-MoE can to cover feature space and make predictions by constructing multiple heterogeneous expert for the single-eye expression recognition task.

## A.6 IMPLEMENTATION DETAILS.

We trained our network for 200 epochs with batch size of 64 on a NVIDIA GeForce RTX 2080Ti GPU. We set event channels $B$ to 3, number of experts $N_e$ to 8 and $TopK$ to 2. We set the number of input frames/tensors $m$ to 2. We implemented MCO-E Net in PyTorch. We trained AdamW with a weight decay 0.001, and the learning rate was set to 0.0003.

## A.7 EVENT CAMERA

Event cameras (Gallego et al., 2019) detect pixel-level changes in scene reflectance with microsecond-level latency and temporal precision, significantly reducing redundant data. These characteristics enable distinct advantages including microsecond response times (¡1 $\mu$s), exceptional dynamic range (up to 140 dB), and energy-efficient operation. Event generation occurs exclusively when logarithmic intensity changes at individual pixels surpass a predefined threshold.

$$\mathcal{E}_i = \{(x_i, y_i, t_i, p_i)\}, i = 1, \ldots, n \tag{18}$$

Here, $\mathcal{E}_i$ denotes the $i$-th event, where $(x_i, y_i)$ represents pixel coordinates, $t_i$ specifies the timestamp with microsecond precision, and $p_i \in \{\pm 1\}$ indicates polarity. The polarity $p$ is determined by the direction of brightness change: $p = +1$ corresponds to brightness increase, and $p = -1$ to brightness decrease at the pixel location.

Following Ahmad et al. (2023), we also convert the asynchronous event data into a voxel grid. The event sequence is represented as $E_1, E_2, \cdots, E_m$, where the dimension of each event tensor

$E_i \in \mathbb{R}^{H \times W \times B}$. In the event tensor $E_i$, the spatio-temporal coordinates, $x_k \in H$, $y_k \in W$, $t_b \in (B-1)$, lie on a voxel grid such that $x_k \in \{1, 2, ..., H\}$, $y_k \in \{1, 2, ..., W\}$, and $t_b \in \{t_0, t_0 + \triangle t, ..., t_0 + (B-1) \triangle t\}$, where $t_0$ is the first time stamp, $\triangle t$ is the bin size, and $B-1$ is the number of temporal bins and $W, H$ are the sensor width and height.

## A.8 STATE SPACE MODEL (SSM)

Mamba (Gu & Dao, 2023) demonstrates superior capability in modeling complex sequential dependencies through its structured State Space Model (SSM) architecture. This innovation renders it particularly effective for long-sequence processing tasks, where conventional Transformer models (Dosovitskiy, 2020; Huang et al., 2024) face limitations due to their quadratic computational complexity. Unlike Transformer-based approaches, Mamba exhibits linear computational complexity scaling with sequence length, offering superior computational efficiency for extended sequences. Therefore, here we introduce the construction process of SSM.

State Space Models (SSMs) (Gu et al., 2021a; Gu & Dao, 2023) are control-theoretic frameworks that formalize dynamical systems for sequential data processing. Originally derived from linear system theory, these models employ state transitions and observation equations to capture temporal dependencies in discrete sequences:

$$
\begin{aligned}
h'(t) &= \mathbf{A}h(t) + \mathbf{B}x(t), \\
y(t) &= \mathbf{C}h(t) + \mathbf{D}x(t).
\end{aligned}
\tag{19}
$$

SSM is defined by four parameters $(\mathbf{A}, \mathbf{B}, \mathbf{C}, \mathbf{D})$, where $\mathbf{A}$ is the state matrix, $\mathbf{B}$ is the input matrix, $\mathbf{C}$ is the output matrix and $\mathbf{D}$ is the feedforward matrix.

**Discretization.** The continuous parameters $A$ and $B$ undergo discretization through transformation methods using the timescale parameter $\Delta$, yielding discrete counterparts $\overline{\mathbf{A}}$ and $\overline{\mathbf{B}}$. This process can be implemented with numerical integration techniques, particularly through the zero-order hold (ZOH) method formalized in equation (20):

$$
\begin{aligned}
\overline{\mathbf{A}} &= \exp(\mathbf{\Delta A}), \\
\overline{\mathbf{B}} &= (\mathbf{\Delta A})^{-1}(\exp(\mathbf{\Delta A}) - \mathbf{I}) \cdot \mathbf{\Delta B}.
\end{aligned}
\tag{20}
$$

The discretization process maps continuous-time parameters $(\Delta, \mathbf{A}, \mathbf{B}, \mathbf{C}, \mathbf{D})$ to their discrete counterparts $(\overline{\mathbf{A}}, \overline{\mathbf{B}}, \mathbf{C}, \mathbf{D})$. Following discretization, the differential equation of SSM can be expressed as follows:

$$
\begin{aligned}
h_t &= \overline{\mathbf{A}}h_{t-1} + \overline{\mathbf{B}}x_t, \\
y_t &= \mathbf{C}h_t + \mathbf{D}x_t.
\end{aligned}
\tag{21}
$$

In addition, the unidirectional SSM designed by Mamba (Gu & Dao, 2023) can only process sequences in one direction, while visual tasks require contextual information. Therefore, many methods (Zhu et al., 2024; Wan et al., 2024; Zhao et al., 2024) use SSM with bidirectional scanning to capture global dependencies and overall semantics more accurately.

## A.9 VISUALIZATION

The color distribution of the heat map reveals the complex relationship between emotional expression and eye dynamic characteristics as shown in Fig. 9. Under normal lighting conditions, anger and disgust show highly similar eyelid edge activation patterns, but the former extends more significantly in the triangular high heat area at the outer corner of the eye, suggesting that the model may capture the subtle boundary between the two emotions. The pupil area of the happiness emotion maintains a stable annular thermal envelope in all types of lighting, and the yellow ring structure around the eye is in sharp contrast to the red pupil of the surprised emotion. It is particularly noteworthy that the heat map of the neutral state shows a unique diffusion characteristic under overexposure conditions. The uniform distribution of light blue around the eyes contrasts with the fragmented response of other emotions in the same lighting, indicating that the model has established an independent lighting invariant representation for the emotion-deficient state.

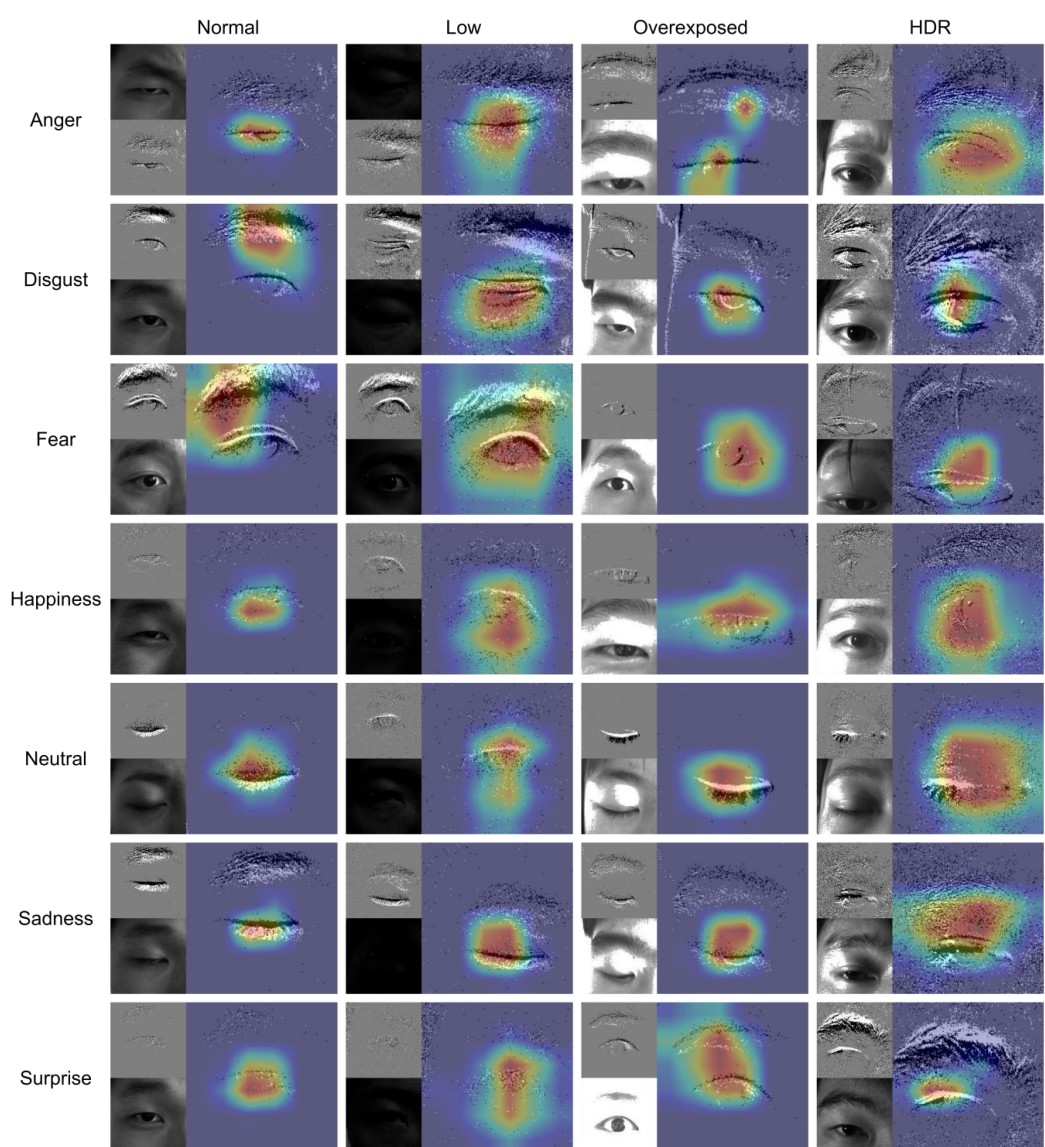

Figure 9: Heat Map Visualization of our proposed MCO-E Net

The thermal shift phenomenon caused by low-light environment reveals the adaptive mechanism of the model. The high response area of the inner corner of the eye for fear and sadness spreads toward the glabella in low light. This migration pattern may reflect the model's compensation strategy for the loss of eye details - maintaining the ability to distinguish emotions by tracking a wider range of muscle group movements. The performance of the HDR channel is particularly prominent. Separate high thermal cores are observed in the middle and lower eyelids in anger, while surprise shows synchronous activation of the pupil center and upper eyelid. This hierarchical response feature suggests that the model adopts differentiated feature fusion strategies under different lighting conditions.

Fig. 9 shows that the model is most robust in encoding happiness and surprise. In the full range of illumination from low light to HDR, the annular thermal structure of happiness remains intact, while the pupil high thermal area of surprise always occupies the visual center. In contrast, the thermal diffusion of the neutral state under overexposure conditions increases compared to normal illumination, but the low-temperature uniform distribution pattern of the periocular muscles remains recognizable, indicating that the model's judgment of the baseline state relies on the overall distribu-

tion characteristics rather than local hot spots. These observations together confirm that the feature fusion mechanism in the design effectively balances local details and global information.

### A.10 LIMITATIONS

While MCO-E Net achieves SOTA performance (WAR and UAR on SEE and DSEE datasets), its computational efficiency presents opportunities for optimization toward edge deployment. The bidirectional state-space modeling in MCO-Mamba necessitates dual-sequence processing for RGB and event modalities, leading to quadratic complexity growth relative to input sequence length. our design contributes to 1.77 GFLOPs, which may challenge resource-constrained devices like AR headsets or embedded systems.

These architectural trade-offs were justified by significant accuracy gains under challenging lighting conditions. Future work could explore hardware-aware neural architecture search to balance computational efficiency and latency.

Notably, our model still outperforms prior works in speed-accuracy trade-offs, and the limitations primarily reflect inherent challenges in fusing long-range spatiotemporal modalities rather than algorithmic deficiencies.

### A.11 STATEMENT

#### A.11.1 ETHICS STATEMENT

We have adhered to the ICLR Code of Ethics in all stages of this research, including paper submission, reviewing, and discussion. Our study does not involve human subjects, and we have complied with all relevant legal and ethical guidelines. There are no conflicts of interest, sponsorship concerns, or biases in our work. We are committed to maintaining research integrity and transparency throughout the process.

#### A.11.2 REPRODUCIBILITY STATEMENT

We have taken steps to ensure the reproducibility of our work. The source code for the models and algorithms used in this paper is available in the supplementary materials. All datasets used in the experiments are described in detail, with processing steps outlined in the appendix. Our methods and results can be independently verified by following the instructions in these materials.

#### A.11.3 THE USE OF LARGE LANGUAGE MODELS (LLMS)

No large language models (LLMs) were utilized in the ideation, writing, or analysis of this research. All conceptualization, research design, data collection, analysis, and manuscript preparation were carried out independently by the authors, without the assistance of any automated language generation tools or AI models. The content of this paper is solely the result of the authors' original work and intellectual contributions.

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
