# OpenReview forum: "Multi-modal Collaborative Optimization and Expansion Network for Event-assisted Single-eye Expression Recognition"
_ICLR.cc/2026/Conference — ICLR 2026 Conference Withdrawn Submission_

### Official Review · Reviewer_56ty · 2025-10-27

**Soundness:** 2
**Presentation:** 2
**Contribution:** 2
**Rating:** 4
**Confidence:** 3

**Summary:**

This paper proposes the Multi-Modal Collaborative Optimization and Expansion Network (MCO-E Net) for event-assisted single-eye expression recognition. The authors address the difficulty of recognizing expressions under extreme lighting conditions by combining RGB and event modalities within a unified collaborative framework. The method contains two main components: (1) MCO-Mamba, a multi-modal collaborative optimization module based on state-space modeling that jointly processes RGB and event streams through bidirectional scanning and shared parameterization; and (2) HCE-MoE, a heterogeneous collaborative and expansion mixture-of-experts that allocates attention to modality-specific experts while reducing overlap through orthogonal regularization.

**Strengths:**

1. Experiments
Experiments on the SEE and DSEE datasets demonstrate that MCO-E Net outperforms existing methods by 2–4% in both WAR and UAR, showing greater robustness under different illumination settings. Ablation studies further verify the individual contributions of MCO-Mamba and HCE-MoE, as well as the influence of expert number and Top-k routing.

2. Architectural design
The two complementary modules (MCO-Mamba for bidirectional SSM-based fusion; HCE-MoE for heterogeneous expert routing with orthogonal regularization) form a coherent framework.

3. Empirical gains
Consistent 2–4% improvements on WAR/UAR over strong baselines on SEE/DSEE; ablations attribute gains to both MCO-Mamba and HCE-MoE.

**Weaknesses:**

1. Generalization scope
Evidence is limited to SEE/DSEE; cross-dataset, cross-subject, and cross-device evaluations (different event sensors/cameras) are not demonstrated.

2. Clarity of components
Some module definitions (e.g., gating reductions, symbol shapes, interaction operators) need explicit dimensionality and implementation details to ensure exact replication.

3. Ablation coverage
Missing or light on key controls such as RGB-only vs Event-only, bidirectional vs unidirectional SSM, router variants, and per-module compute/latency trade-offs.

4. Robustness and stability
The results show moderate improvements (around 2–4%) over existing methods, which are not particularly large given the added architectural complexity. The absence of variance or standard deviation reporting raises concerns about performance fluctuation and statistical reliability across multiple runs.

**Questions:**

1. The manuscript uses both “Topk” and “Top-K”. Please unify as “Top-k” (or “Top-K”) consistently across the paper.

2. Line 409: “2 type of experts” should be “2 types of experts”.

3. Line 860: “33.3%0.5%” is missing the ± sign; it should be “33.3% ± 0.5%”.

---

### Official Review · Reviewer_iv73 · 2025-10-30

**Soundness:** 2
**Presentation:** 2
**Contribution:** 1
**Rating:** 2
**Confidence:** 5

**Summary:**

This paper presents MCO-E Net, a multimodal framework for event-assisted single-eye expression recognition that integrates RGB and event data through two novel modules: MCO-Mamba, which enables cross-modal joint optimization and semantic interaction based on the Mamba architecture, and HCE-MoE, a heterogeneous mixture-of-experts module combining deep, attention, and focal experts via an attention-guided router. The proposed model achieves state-of-the-art results on SEE and DSEE datasets, demonstrating strong robustness under challenging lighting conditions and fast inference speed.

**Strengths:**

1. The combination of Mamba and heterogeneous MoE for multimodal feature fusion is creative and technically interesting.
2. Experimental results are strong, showing consistent improvements over state-of-the-art methods across all lighting conditions with very low inference latency.
3. The motivation is clear and addresses a real challenge of semantic misalignment between RGB and event modalities.
4. Ablation studies are comprehensive, verifying the contribution of each component and parameter choice.

**Weaknesses:**

1. Although the model is complex from an engineering perspective, there is limited theoretical analysis of the convergence and dynamic equilibrium mechanisms of the MCO-Mamba joint optimization.
2. Mamba-based multimodal fusion shares similarities with recent approaches (e.g., Sigma, MSFMamba, DepMamba, 2024–2025), where the innovations primarily arise from module combinations rather than fundamentally new principles.
3. The validation is limited to two similar monocular datasets, without cross-dataset or real-world application testing, which makes it difficult to assess the generalization capability.
4. Although the tables are comprehensive, statistical significance metrics (e.g., standard deviation or p-value) are not reported, suggesting that some improvements may fall within the noise range.
5. Writing and notation clarity could be improved, with some overly long sections and insufficient explanation of key equations.

**Questions:**

1.How does the joint optimization function M(A,B) ensure stable gradients and avoid dominance by one modality?
2.What is the computational overhead distribution among the three expert types in HCE-MoE? Any observed load imbalance or expert collapse?
3.Have you tested the model’s transferability to other multimodal tasks (e.g., facial emotion recognition, driver monitoring)?
4.Are training hyperparameters and hardware configurations fully reproducible from the released code repository?

---

### Official Review · Reviewer_hzxH · 2025-10-30

**Soundness:** 2
**Presentation:** 2
**Contribution:** 2
**Rating:** 2
**Confidence:** 5

**Summary:**

The paper proposes MCO-E Net, a multimodal architecture for recognizing single-eye expressions that combines RGB and event camera data. It introduces two components: (1) MCO-Mamba: a variant of Mamba featuring a joint optimization scheme (MJOS) and a collaborative interaction block (MCIB) for cross-modal fusion; and (2) HCE-MoE: a heterogeneous mixture-of-experts module comprising structurally diverse experts, as well as an attention-enhanced router. The method was evaluated using the SEE and DSEE corpora and achieved competitive WAR/UAR scores.

**Strengths:**

1) The use of event cameras for illumination-robust eye expression recognition is motivated by practical considerations.
2) The paper includes component and hyperparameter ablations.
3) Despite model complexity, efficiency analysis shows reasonable speed.

**Weaknesses:**

1) The experiments were conducted on two corpora only (SEE and DSEE). The method's practical applicability is called into question by the lack of verification on independent, in-the-wild corpora.
2) No experiment tests the transferability of the model (e.g. training on SEE and testing on DSEE, or on a third-party corpus). The latter is more important.
3) The MJOS proposal essentially involves jointly projecting parameters from two modalities, followed by residual addition. This is technically similar to existing methods, such as those that also use joint updating of SSM parameters. It is not a novel concept that should be presented and promoted.
4) The use of multiple types of experts (Deep, Attention and Focal) is an empirical ensemble rather than theoretically justified heterogeneity. It is unclear why these three types in particular were chosen, or why their combination is fundamentally better than a combination of four or two types.
5) The justification for the complexity of the MCO-E Net is unclear.
6) As there is no information about the dispersion of metrics, the advantage is statistically insignificant.
7) How exactly were the frames and events synchronized? Was hardware synchronization or post-processing used?
8) There are no specific examples of incorrect predictions.

**Questions:**

1) Could you demonstrate the performance of the model on a third, independent corpus? Ideally, this should be one with real-world lighting variations, motion blur or cross-subject splits, in order to validate generalizability.
2) How exactly were the frames and events synchronized?
3) Was hardware synchronization or post-processing used?

---

### Official Review · Reviewer_XWvW · 2025-11-02

**Soundness:** 3
**Presentation:** 2
**Contribution:** 2
**Rating:** 4
**Confidence:** 4

**Summary:**

This paper leverages event modalities to resist challenges such as low light, high exposure, and high dynamic range in single-eye expression recognition tasks. To this end, the authors propose a Multi-modal Collaborative Optimization and Expansion Network (MCO-E Net) to facilitate RGB-Event fusion and collaboration.

**Strengths:**

The MCO-E Net achieves competitive performance on the single-eye expression recognition task while attaining the fastest inference speed as shown in Section 4.1 and A.3.

**Weaknesses:**

1. It would be helpful to clarify the necessity and significance of incorporating event-based data specifically for the eye-related facial expression recognition task. Additionally, it should be explicitly stated which aspects of the method are specifically designed to cater to the challenges of eye-related facial expression recognition.

2. The proposed network structure lacks significant innovation. For instance, in the introduction of MCO-Mamba, MJOS derives the BC parameters of SSM by concatenating RGB and event features, which is similar to existing approaches (such as Sigma), where the C parameters of SSM are obtained through features from other modalities. Furthermore, the distinction between the router in HCE-MoE and existing attention routers is minimal, as it mainly involves the addition of two linear layers.

3.The proposed method is a multimodal approach, yet it has not been compared with existing multimodal methods. The comparison is limited to monocular recognition methods. It would be valuable to also compare the approach with current multimodal methods, as they may also be applicable to RGB-Event fusion.

**Questions:**

1. It is suggested to analyze how the different experts (Router with Attention, Deep Expert, Attention Expert, and Focal Expert) contribute to eye-related emotion recognition. Specifically, why were these particular experts chosen, and how does each one enhance the model's performance for this specific task?

2. Since event data is difficult to obtain in most cases, could this method be extended to recognize eye emotions using only an RGB dataset? For example, could the event data be simulated based on the provided RGB dataset?

---

### Note · Authors · 2025-11-12

I have read and agree with the venue's withdrawal policy on behalf of myself and my co-authors.